# Chemical Profiling and Therapeutic Evaluation of Standardized Hydroalcoholic Extracts of *Terminalia chebula* Fruits Collected from Different Locations in Manipur against Colorectal Cancer

**DOI:** 10.3390/molecules28072901

**Published:** 2023-03-23

**Authors:** Soibam Thoithoisana Devi, Khaidem Devika Chanu, Nameirakpam Bunindro Singh, Sushil Kumar Chaudhary, Ojit Singh Keithellakpam, Kshetrimayum Birla Singh, Pulok K. Mukherjee, Nanaocha Sharma

**Affiliations:** 1Institute of Bioresources and Sustainable Development (An Autonomous Institute under the Department of Biotechnology, Goverment of India), Takyelpat, Imphal 795001, Manipur, India; 2Department of Zoology, Manipur University (MU), Imphal 795003, Manipur, India; 3School of Biotechnology, Kalinga Institute of Industrial Technology (KIIT), Deemed to be University, Bhubaneshwar 751024, Odisha, India

**Keywords:** *Terminalia chebula* Retz., north east India, GC-MS, HPTLC, cytotoxicity, HCT 116

## Abstract

*Terminalia chebula* Retz. (Fam. Combretaceae), locally called Manahei, is a well-known medicinal plant that grows wildly in Manipur, a Northeastern state of India. It is used as a mild laxative, an anti-inflammatory agent, and a remedy for piles, colds, and ulcers by ethnic communities of the state. The hydroalcoholic extract obtained from four fruit samples of *T. chebula* collected from different locations in Manipur were analyzed using gas chromatography–mass spectrometry (GC-MS) and high-performance thin-layer chromatography (HPTLC) for their chemical constituents and evaluated for their anticancer activity against the colon cancer cell HCT 116. GC-MS analysis results indicated significant variation in the composition and percentage of major compounds present in the extracts. 1,2,3-Benzenetriol was the most abundant chemical constituent present in all four extracts of *T. chebula*, ranging from 20.95 to 43.56%. 2-Cyclopenten-1-one, 5-hydroxymethylfurfural, and catechol were commonly present in all extracts. Two marker compounds, gallic acid and ellagic acid, were also quantified usingHPTLC in all four extracts of *T. chebula*. The highest content of gallic acid (22.44 ± 0.056 µg/mg of dried extract) was observed in TCH, and that of ellagic acidwas found in TYH (11.265 ± 0.089 µg/mg of dried extract). The IC_50_ value of TYH for the DPPH and ABTS assays (12.16 ± 0.42 and 7.80 ± 0.23 µg/mL) was found to be even lower than that of Trolox (18 ± 0.44 and 10.15 ± 0.24 µg/mL), indicating its strong antioxidant properties among the four extracts of *T. chebula*. The MTT assay determined the effect of *T. chebula* extracts on the viability of HCT 116 cells. TYH showed the highest activity with anIC_50_ value of 52.42 ± 0.87 µg/mL, while the lowest activity was observed in TCH (172.05 ± 2.0 µg/mL). The LDH assay confirmed the cytotoxic effect of TYH in HCT 116 cells. TYH was also found to induce caspase-dependent apoptosis in HCT 116 cells after 48 h of treatment. Our study provides insight into the diversity of *T. chebula* in Manipur and its potential activity against colon cancer.

## 1. Introduction

*Terminalia chebula* Retz. (Fam. Combretaceae) is a species of *Terminalia* that is commonly known as black, yellow, or chebulic myrobalan. It is a flowering deciduous tree that grows to a height of around 30 m. It thrives in places with altitudes of up to 2000 m above sea level, 100–150 cm of annual rainfall, and temperatures between 0 and 17 °C [1]. This medicinal plant is indigenous to South and Southeast Asia, where it grows naturally in countries including India, Sri Lanka, Bhutan, Nepal, Bangladesh, Myanmar, Cambodia, Laos, Vietnam, Indonesia, Malaysia, Pakistan, and Thailand. It grows in Fujian, Guangdong, Guangxi (Nanning), and Taiwan and is native to Yunnan in China (Nantou). It is also widely found in Iran, Afghanistan, Africa, and Brazil [2,3,4,5]. *T. chebula* is known as the “King of Medicine” in Tibet. Because it can treat a wide range of illnesses and is revered by God Siva, it is known as “Haritaki” (Hara) [6].

*T. chebula* has long been used in Ayurveda, Unani, and Homeopathic medicine and has become one of the most explored species of the genus *Terminalia* [7]. *T. chebula* fruits have astringent, carminative, purgative, and stomachic properties [8,9]. In traditional Ayurvedic and Siddha medicine, fruits are used to treat conditions such aschronic diarrhea, gastroenteritis, constipation, malabsorption syndrome, asthma, ulcer, dyspnea, dyspepsia, hemorrhoids, cough, candida infection, antiparasitic, urinary discharge, antitumor, skin disorders, loss of memory, epilepsy, cardiovascular problems, diabetes, lack of appetite, and homeostatic dysfunction [10,11,12]. *T. chebula* is present in Triphala rasayana, an ayurvedic formulation comprising fruits of three plant species, viz., *Phyllanthus emblica* L. (*P. emblica*), *Terminalia chebula* Retz. (*T. chebula*), and *Terminalia bellirica* Roxb (*T. bellirica*). It has a great deal of potential to enhance the gastrointestinal tract (GIT) system and is particularly helpful against gastrointestinal disorders such as constipation, gastric ulcers, and inflammatory bowel disease (IBD), among others [13]. The ethnic populations of Manipur traditionally use it as a light purgative, an anti-inflammatory, and a remedy for piles, colds, and ulcers [14]. The Meitei community of Manipur has also been reportedly using the fruits of *T. chebula* for the treatment of hepatic disorder [15].

Manipur is a northeastern state of India bordering Myanmar. The flora of Manipur falls into two biodiversity hotspots, the Himalayan and the Indo-Burma global biodiversity hotspot [16]. It has an extensive germplasm of medicinal plants, but very few works have been reported on their chemical nature and biological functions. Different phytoconstituents are found in plants, and these vary depending on environmental factors (humidity, rainfall, temperature) and geographic region (latitude, soil duration of sunlight) [17]. The biological activity of a medicinal plant is mainly determined by its chemical constituents, which are affected by many elements including environmental conditions [18]. The diverse pharmacological activities of *T. chebula* may be attributed to several bioactive phytoconstituents, including phenols, flavonoids, tannins, saponins, sterols, amino acids, etc., with tannins being the major phytocomponents present, accounting for 32% of the phytochemicals present. The tannin concentration of *T. chebula* has also reportedly been linked to variations in geographic conditions. Chebulic acid, chebulinic acid, chebulagic acid, gallic acid, corilagin, and ellagic acid make up the majority of the tannins in *T. chebula* [19]. The fruit extract of *T. chebula* from northeast India, especially Manipur, has not been explored for its phytochemical composition and anticancer properties.

Colorectal cancers are the most common gastrointestinal malignancies in the world. According to the 2018 Globocan report, this contributes to 9.2% of all major cancer incidences worldwide [20]. Colon cancer ranks eighth and rectal cancer ranks ninth among the most commonly diagnosed cancers in India [21,22]. The incidence of colorectal cancer has been found to be highest in the northeast regions of India. Dietary factors such asconsumption of beef, pungent spices, and red meat are believed to have contributed to the higher incidences of CRC in the region [23,24,25]. According to numerous studies, eating fruits and vegetables, which include fiber, antioxidants, and various vitamins and minerals with anticancer qualities, can reduce the incidence of colorectal cancer [26,27,28,29]. Secondary metabolites such asflavonoids, phenolics, terpenoids, saponins, quinines, and alkaloids have been proven to show chemoprotective activity against CRC [30].

Our study is aimed atexploring the phytochemical profile of the hydroalcoholic extract of *T. chebula* fruits collected from different locations in Manipur, India, and to evaluate its anticancer activity against the colon cancer cell HCT 116. GC-MS was used for identifying the volatile constituents of the extracts and quantifying the relative abundance of each constituent in the extract. HPTLC analysis is a technique that is commonly used for the validation of traditional medicines derived from plants. Two major hydrolyzable tannins, gallic and ellagic acid, were also quantified from the fruit extracts of *T. chebula* using the HPTLC, DPPH, and ABTS assays to measure the extracts’ antioxidant properties. The anticancer activity of *T. chebula* extract was studied in HCT 116 colon cancer cells, and the mechanism for cell cytotoxicity was studied using various assays.

## 2. Results

### 2.1. Yield of the Extract

The percentage yield of the extracts of *T. chebula* collected from different locations in Manipur is given in Table 1. Qualitative phytochemical screening revealed the presence of tannins, saponins, flavonoids, alkaloids, phenolics, acid, and glycosides.

### 2.2. GC-MS Analysis of Hydroalcoholic Extracts of T. chebula Fruits

The results of the GC-MS analysis led to the identification of several compounds from the extracts of *T. chebula* collected from four different locations in Manipurthrough the National Institute of Standards and Technology (NIST) GC-MS libraries. The GC-MS analysis of the extracts led to the identification of 71, 64, 59, and 63 compounds in TCH, TKH, TSH, and TYH. Appendix A show the phytocompounds detected, retention time (RT), RSI value, molecular weight, and area of the peak concentration(%). All four chromatograms are also displayed in Appendix A. Predominant compounds detected in the extracts consist of1,2,3-benzenetriol, 2-cyclopenten-1-one, 5-hydroxymethylfurfural, and catechol. Appendix A provides structural representation of major compounds detected in the *T. chebula* extracts. Other major compounds include phenol, 2,5-furandione, 1-hexyl-2-nitrocyclohexane, 1-undecanol, hexadecanamide, sedanolide, 4H-pyran-4-one, etc. Table 2 depicts a comparative study of the major chemical constituents present in the four fruit extracts of *T. chebula*. Most of the compounds detected are reported to exhibit important biological activities. It is evident from the results obtained from the GC-MS analysis that variation in the composition of phytochemicals was observed between the different extracts of *T. chebula*. The percentage composition of the major compounds present was also found to vary among the different extracts. In all four extracts, 1,2,3-benzenetriol was found to be the most prevalent chemical component, with concentrations ranging from 20.95% (TCH) to 43.56%. (TYH). 2-Cyclopenten-1-one was the second most abundant chemical constituent, ranging from 9.07% (TYH) to 19.35% (TCH). The GC-MS chromatogram of TCH yielded seven major peaks, which included1,2,3-benzenetriol (20.95%), 2-cyclopenten-1-one (19.35%), catechol (12.82%), 5-hydroxymethylfurfural (6.04%), 1-hexyl-2-nitrocyclohexane (4.13%), 1-undecanol (3.15%), and phenol (3.27%). TSH yielded fivemajor peaks, whichincluded 1,2,3-benzenetriol (40.41%), 2-cyclopenten-1-one (14.66%), catechol (12.47%), phosphonic acid (4.64%), and 1-hexyl-2-nitrocyclohexane (2.11%). Eightmajor peaks were detected in TKH, which included 1,2,3-benzenetriol (26.53%), 2-cyclopenten-1-one (12.79%), 5-hydroxymethylfurfural (11.26%), catechol (6.85%), phosphonic acid (3.85%), 2,6-diflourobenzoic acid (3.05%), 1-hexyl-2-nitrocyclohexane (2.76%), and sucrose (2.27%). Fourmajor peaks were detected in TYH, which included the compounds 1,2,3-benzenetriol (43.56%), 2-cyclopenten-1-one (9.07%), phenol (5.40%), and 5-hydroxymethylfurfural (4.03%). The GC-MS analysis of each *T. chebula* fruit extract revealed significant variations in both the quantity and nature of the chemical components. Active substances present in plants belonging to the same species may differ in their type, quantity, and constituent proportion depending on numerous factors, including internal genetic variability and external environmental factors (light, temperature, water, salinity, latitude, etc.) [31].

### 2.3. High-Performance Thin-Layer Chromatography (HPTLC)

In the study, the quantification of gallic acid and ellagic acid in the hydroalcoholic extracts of *T. chebula* collected from different locations in Manipur isdisplayed in the form of the HPTLC profile in Figure 1. The use of toluene–ethylacetate–formic acid–methanol in the ratio of 6:3:0.1:0.9 and toluene–ethylacetate–formic acid in the ratio of 7:3:1 provided compact spots for gallic acid and ellagic acid, respectively, at R*f* = 0.220 and R*f* = 0.187, as shown in Figure 1A,B. Anisaldehyde solution was used to derivatize the plates and further heated at 100 °C for 3 min. The plates were then examined underwhite light and at 366 nm. The linear regression line equation of gallic acid and ellagic acid derived and the correlation coefficient (R^2^)was found to be 0.99991 and 0.999681, respectively. The percentages of the two marker compounds in the hydroalcoholic extract of *T. chebula* are listed in Table 1. Gallic and ellagic acid content was found to vary among the different extracts of *T. chebula*. Among the extracts, the highest quantity of gallic acid was obtained in TCH (22.44 ± 0.056 µg/mg of dried extract), while that of ellagic acid was found to be highest in TYH (11.265 ± 0.089 µg/mg of dried extract).

### 2.4. Total Phenolic, Total Flavonoid, and Antioxidant Activity of T. chebula Extracts

In order to calculate the quantity of polyphenols and flavonoids present in the extracts, gallic acid and quercetin standards were used to derive the calibration curve, y = 0.010x + 0.251, R2 = 0.983 and y = 0.016x − 0.096, R2 = 0.963, for the determination of TPC and TFC, respectively. TPC and TFC were expressed in terms of gallic acid equivalent and quercetin equivalent per gram of dry weight of extract, respectively. TPC was found to be highest in TYH (304.56 ± 1.23 GAEmg/g ext), while the highest TFC was observed in TCH (8.98 ± 0.35 QEmg/g ext). The free radical scavenging or antioxidant capacity of the four *T. chebula* extracts was assessed using the DPPH and ABTS assays. The results were compared with Trolox (TC) and ascorbic acid (AC) standards. Figure 2 depicts the IC_50_ values (concentration of extracts that cause 50% of free radical scavenging activity) for the DPPH and ABTS assaysand TPC and TFC of the extracts. TYH was found to possess the highest activity in both DPPH and ABTS assays with the most negligible IC_50_ value, as shown in Table 3. The highest concentration, 500 µg/mL, of TYH showed 85% free radical scavenging activity in the DPPH assay, and 100 µg/mL of TYH showed 90% activity in the ABTS assay. The IC_50_ value of TYH for the DPPH and ABTS assays (12.16 ± 0.42 and 7.80 ± 0.23 µg/mL) was found to be lower than that of Trolox (18 ± 0.44 and 10.15 ± 0.24 µg/mL). The free radical scavenging activity was in the order TCH < TSH < TKH < TYH < TC < AC.

### 2.5. MTT Assay

Cell viability represents the number of healthy cells present in a given population [32]. A prominent indicator of cellular health is metabolic activity. Cell-based colorimetric assays such as the MTT assay are often used to determine cellular metabolic activity [33]. It is used as an indicator for cell viability, proliferation, and cytotoxicity. The invitro MTT assay was performed to study the effect of *T. chebula* extracts on the viability of HCT 116 cells and its cytotoxic effect. Cells were treated with increasing concentrations of *T. chebula* (0–500 µg/mL) for 48 h, and viability was determined using MTT (3-(4,5-dimethylthiazol-2-yl)-2,5-diphenyl tetrazolium bromide) assay. The IC_50_ values of all four extracts are shown in Table 4. A dose-dependent effect was observed for all extracts, as shown in Figure 3B. TYH showed the highest activity against HCT116 cells with an IC_50_ value of 52.42 ± 0.87 µg/mL, comparable to standard drugs 5-FU and cisplatin (Figure 3C). A drastic effect on the viability of HCT 116 cells was observed in the case of treatment with TYH. At the highest concentration (500 µg/mL) of TYH treatment, the percentage viability was as low as 26%. The observation of cell morphology after treatment with *T. chebula* extracts for 48 h showed a significant change in cell morphology and a reduction in cell number (Figure 3A). Compared to control cells, the TYH-treated cells showed a significant change in cell morphology, including loss of membrane integrity, loss of contact from neighboring cells, detachment from the culture plate, and cellular condensation.

### 2.6. Effect of T. chebula Extracts on Colony-Forming Capacity of HCT 116 Cells

As evident from the above results, TYH showed the highest anticancer activity against HCT116. Further studies to validate the effect of the hydroalcoholic extract of *T*. *chebula* found in Manipur were carried out using TYH. Cellular proliferation is the capacity of healthy cells to create progeny and divide [34]. The antiproliferative activities of the extract were further evaluated using thecolony-forming assay. Theclonogenic assay, also known as a colony-formation assay, is an in vitro cell survival assay based on the capacity of a single cell to develop into a colony. It essentially tests the ability of cells in a population to undergo unlimited cell division [35]. Hence, this assay directly elucidates the capacity of a therapeutic to inhibit tumor proliferation. Figure 4A,B shows that the colony formation decreases as the extract concentration increases. TYH showed the highest antiproliferative activity, with only 5% colony formation at 20 µg/mL.

### 2.7. Cytotoxic Activity of TYH on HCT 116 Colon Cancer Cell

Using the CyQuant LDH Cytotoxicity Assay Kit (Invitrogen, Eugene, OR, USA), the LDH assay was performed to confirm the cytotoxic impact of TYH on HCT 116 cells. Since LDH is a cytosolic enzyme thatis released upon membrane damage, the level of LDH enzyme in the cytoplasm is directly proportional to cytotoxicity. The control cells (non-treated) showed very lowrelease in LDH compared to cells treated with TYH. As shown in Figure 5, there was asignificant increase in the level of LDH in a dose-dependent manner with increasing concentrationsof TYH. The release of intracellular LDH in the medium is a measure of theirreversible death of cells due to cellular damage. It has also been also reported that LDH upregulation leads to subsequent induction of apoptosis.

### 2.8. Effect of TYH on Migration of HCT 116 Cells

The scratch wound assay was performed to determine whether TYH affects the migration of HCT 116 cells. An increasing dose (non-cytotoxic) of TYH was used to treat the HCT 116 cells. A 5 µg/mL (non-lethal) amount of 5-FU was used as the positive control in this experiment. Photographs of the scratched area were taken every 0, 24, and 48 h. As shown in Figure 6, it was discovered that TYH inhibited HCT 116 cells’ migration in a dose- and time-dependent manner. At 48 h of treatment, the cells in the untreated well had migrated and covered the scratched area, whereas lesser migration was observed in the treated ones.

### 2.9. Analyses for Apoptosis

#### 2.9.1. Effect of TYH on the Morphology of HCT 116 Cells

HCT 116 cells were stained with 6% Giemsa, and the change in morphology of the nucleus and cytoplasm was examined after 48 h of treatment with TYH. As seen in Figure 7A. The control group has a much higher number of cells than the other treatment groups. Shrinkage and cellular damage become more prominent with the increase in concentration (60 and 90 µg/mL) of the extract. At 120 µg/mL, almost no normal cell morphology could be seen.

#### 2.9.2. Apoptotic Effect of TYH Observed through DAPI Staining

Changes in nuclear morphology induced due to apoptosis in HCT 116 cells were observed with DAPI staining. A dose-dependent pattern of nuclear morphological alterations indicative of apoptosis was seen in cells treated with TYH. As shown in Figure 7B, the cells in the control group have intact and evenly colored nuclei, and the cells do not show characteristics of apoptosis. With the increase in treatment concentration, nuclear condensation and apoptotic body formation were visible. At 120 µg/mL, the number of cells is drastically decreased, accompanied by increased nuclear condensation and fragmentation.

#### 2.9.3. TYH Activates Caspase-3 in HCT 116 Cells

TheEnzChek^®^ Caspase-3 Assay Kit detected apoptosis by assaying the increase in caspase-3 and other DEVD-specific protease activity. Using this fluorometric assay, we determined whether the incubation of HCT 116 cells with TYH for 48 h could induce caspase-3 activity. Compared to the control group, TYH significantly activated caspase-3. With the increase in the concentration of TYH, the activity of caspase-3 was found to increase in a dose-dependent manner with respect to time (30–120 m). The amount of AMC (nM) released also escalated with an increase in THY concentration at 120 min, as shown in Figure 7C,D.

## 3. Discussion

In this study, we demonstrated a comprehensive profile of the diversity in the chemical composition of the hydroalcoholic extract of the fruits of *T. chebula* found in Manipur, a northeastern state of India through GC-MS and HPTLC analyses. The cytotoxic and antiproliferative activities of the extract were also studied against the colorectal cancer cell line HCT 116.

The yield of the hydroalcoholic extract of *T. chebula* fruits was found to be in the range of 25.2–30.3% (*w*/*w*) Table 1. The maximum yield of the extract was found in TKH (30.3%), followed by TYH and TSH, with a yield % of 28.7%. The results are in accordance with previous works conductedon *T. chebula* but with different solvent systems used for extraction [36]. GC-MS analysis of the individual extracts led to the identification of significant compounds in accordance with previous reports of exerting antioxidant and cytotoxic potential. 1,2,3-Benzenetriol and 2-cyclopenten-1-one are two major compounds detected in the GC-MS analysis and are present in all four *T. chebula.* The high content of 1,2,3-benzenetriol and 2cCyclopenten-1-one in the fruit extract of *T. chebula* has also been reported in previous studies [37,38]. Other major compounds include 5-hydroxymethylfurfural, catechol, phenol, 1-hexyl-2-nitrocyclohexane, 1-undecanol, hexadecanamide, 4H-pyran-4-one, etc. As shown in Table 2, the GC-MS resultsshowedsignificant variation both quantitatively and in terms of chemical composition among the four extracts of *T. chebula* collected from different locations in Manipur. Among the many compounds detected in the extracts of *T. chebula*, some compounds have vast medicinal properties. TYH was found to contain the highest percentage, 43.56%, of 1,2,3-benzenetriol (pyrogallol) compared to other extracts. The percentage content of 1,2,3-benzenetriol in TCH, TKH, and TSH was found to be 20.95%, 26.53%, and 40.41%.1,2,3-Benzenetriol, also known as 1,2,3-trihydroxy benzene or pyrogallol, falls under the category of tannins (hydrolyzable). It exhibits many well-known biological qualities, including those that are antibacterial, antioxidant, antifungal, antiviral, antiseptic, antidermatitic, cardioprotective, antimutagenic, and pesticide, among others [39,40,41,42]. The strong antioxidant properties of pyrogallol have been attributed to the presence of hydroxyl groups in the ortho position of its B ring [43]. Pyrogallol has been found toinhibit the proliferation of human tumor cell lines such asK562, Jurkat, Rajij, and HEL. It has also demonstrated the dose-dependent suppression of SNU-484 gastric cancer cells by causing apoptosis [44,45]. 2-Cyclopenten-1-one and 2-hydroxy-3,4-dimethylalso exhibited antioxidant ability [46]. For MCF-7 breast cancer cells, 2-cyclopenten-1-one functions as a differentiation agent. It causes cell cycle arrest in the G1phase by decreasing the expression of the cyclin D1 gene in these cells. A further study found that cyclin D1 promoter activity is inhibited by 2-cyclopenten-1-one [47].5-Hydroxymethylfurfural has been reported for its antioxidant and antiproliferative activities in human melanoma A375 cells and has great potential application in cancer chemoprevention. The antifungal, anti-inflammatory, antiviral, and antioxidant activities of catechol have been reported in previous studies [48,49,50,51,52].1-Hexyl-2-nitrocyclohexane, hexadecanamide, and 4H-pyran-4-one have also been reported to havea number of biological activities [53,54,55]. Moreover, benzeneacetaldehyde (1.56%), 2,3,4,5-tetrahydropyridazine (1.65%), 2,5-furandione, and 3-methyl(1.18%) were some new active constituents found in TYH. Sedanolide (1.14%), vanillin (0.74%), and isothiazole (1.93%) were also new active constituents found in TKH, TSH, and TCH respectively.

The fruits of *T. chebula* are rich in tannins and mostly hydrolyzable [1]. Examples of hydrolyzable tannins found in *T. chebula* include gallic acid, chebulagic acid, punicalagin, chebulanin, corilagin, neochebulinic acid, ellagic acid, chebulinic acid, 1,6-di-o-galloyl-D-glucose, casuarinin, 3,4,6-tri-o-glloyl-D-glucose, and terchebulin. Gallic acid was found to have the highest antioxidant capacity among many polyphenols [56]. It has been found to significantly suppress the proliferation of HCT 116 and HT-29 colon cancer cells and increase the apoptosis rate. It inhibited the phosphorylation of SRC and EGFR phosphorylation. Ellagic acid has been proven to prevent the growth of HCT 116 by regulating many targets and modulating critical signaling pathways [57]. We quantified gallic acid and ellagic acid, two prominent hydrolyzable tannins (derivatives of pyrogallol), in the hydroalcoholic extracts using HPTLC. As evident from the results of the HPTLC analysis, gallic acid and ellagic content were also found to vary significantly among the different extracts. The highest GA content was seen in TCH (22.44 ± 0.035 µg/mg dry weight of the extract), while the highest ellagic acid content was seen in TYH (11.26 ± 0.07 µg/mg dry weight of the extract). The gallic acid content of the four extracts was in the order TSH < TYH < TKH < TCH, while ellagic acid content was in the order TCH < TKH < TSH < TYH.

The antioxidant capacities of the extracts were determined by two standard assays, DPPH and ABTS. Trolox and ascorbic acid were used as standards. With the lowest IC_50_ values (12.16 ± 0.42 and 7.80 ± 0.23 µg/mL), TYH showed the highest free radical scavenging activity in both assays comparable to that of Trolox and ascorbic acid. The increased antioxidant ability in TYH could be due to the presence of the high amount of phenolic compounds present in it, as evident from the results obtained fromthe GC-MS and HPTLC analyses [58]. Our results also indicated a positive correlation between the antioxidant activity and TPC of the extracts. The correlation had been studied in previous reports also [59,60,61].

Considering the vastmedicinal properties of *T. chebula* and its usage in traditional medicine, the effect of the hydroalcoholic extract of fruits of *T. chebula* collected from different locations in Manipur in HCT 116 colon cancer cells was studied. Thus, using the cytotoxicity assay (MTT assay), we found that the hydroalcoholic extract of *T. chebula* collected from YumnamHuidrom, Manipur (TYH), exerted the strongest anticancer activity in vitro among the four extracts. The IC_50_ values for TCH, TKH, TYH, and TSH were found to be 172.05 ± 2.0 µg/mL, 94.07 ± 1.70 µg/mL, 52.42 ± 0.87 µg/mL, and 133.58 ± 2.0 µg/mL. The four classifications of extracts are very active (IC_50_ ≤ 20 g/mL), moderately active (IC_50_ > 20–100 g/mL), weakly active (IC_50_ > 100–1000 g/mL), and inactive (IC_50_ > 1000 g/mL) [62,63]. A pure chemical or medication is deemed powerful if its IC_50_ value is less than 4 g/mL. TKH and TYH were found to be moderately active and thus can be considered for novel drug development. A satisfactory result was obtained for TYH with its IC_50_ value against HCT 116 comparable to that of standard drugs cisplatin and 5-FU. The mechanism of cytotoxicity and antiproliferation was further investigated by studying the cell morphology using Giemsa and DAPI staining and caspase-3 enzyme activity assay. Results obtained from Giemsa and DAPI indicated the apoptotic activity of TYH in HCT 116 cells. Further validation was provided by the caspase-3 activity assay. Our results indicated that TYH mediated apoptotic cell death in HCT 116 cells.

Our findings revealed the promising anticancer potential of the hydroalcoholic extract of *T. chebula* fruit found in Manipur, a part of thenortheastern region of India, against CRC. This study also provided insight into the diversity of *T. chebula* plants found in the state and the variation in their phytochemical composition. The medicinal and pharmacological properties of this plant and its parts areyet to be explored for its anticancer potential.

## 4. Materials and Methods

### 4.1. Chemical Reagents

DMEM (Dulbecco’s Modified Eagle Medium, Thermo Fisher Scientific, Waltham, MA, USA), Pen Strep (penicillin–streptomycin), PBS (phosphate-buffered saline) and trypsin from Gibco (New York, NY, USA), and FBS (fetal bovine serum) were purchased from Gibco (UK). MTT (3-(4,5-dimethylthiazol-2-yl)-2,5-diphenyltetrazolium bromide) was purchased from Thermo Fisher Scientific (Waltham, MA, USA). DMSO (dimethyl sulfoxide) and methanol were purchased from Rankem (India). ABTS, gallic acid (≥99%), 5-FU (5-fluorouracil) (≥99%), cisplatin, and ellagic acid (≥95%) were obtained from Sigma (St. Louis, MO, USA). Ethanol, DPPH (1,1-diphenyl-2-picryhydrazyl), crystal violet, and Giemsa stain were purchased from Himedia (Mumbai, India). DAPI (4′,6-diamidino-2-phenylindole, dihydrochloride), Catalog No. D1306 (Eugene, OR, USA); EnzChek™ Caspase-3 Activity Assay Kits, Catalog No. E13183 (Eugene, OR, USA); and CyQuant LDH Cytotoxicity Assay Kit, Catalog No.C20300 (Eugene, OR, USA), were purchased from Invitrogen, Thermo Fisher Scientific (Waltham, MA, USA). HCT-116 cells were obtained from NCCS, Pune, as a kind gift from Dr. Manoj Kumar Bhat.

### 4.2. Collection of Samples

The fruits of *T. chebula* (Appendix A) required for the proposed study were collected in the month of November 2021 from four different locations in Manipur. The plant samples were identified and authenticated as *Terminalia chebula* (Manahei), and voucher specimens were submitted at the Plant Systematic and Conservation Laboratory, IBSD, Imphal. Herbaria were submitted, and voucher numbers were obtained as shown in Table 5. Fruit samples with coordinates were Chandel, Manipur ‘TCH’ (N 24°25′35.7″ E94°00′47.7″); Kakching, Manipur ‘TKH’ (N 24°30′01.9″ E93°58′27.1″), Sagolband, Imphal, Manipur ‘TSH’ (N 24°48′06″ E93°55′16′); and YumnamHuidrom, Imphal, Manipur ‘TYH’ (N 24°39′46″ E93°54′14″). The fruits collected were washed with tap water to eliminate any unwanted impurities and dried.

### 4.3. Preparation of Extracts

The dried fruits of *T. chebula* collected from different locations in Manipur were ground into fine powder form. The powdered plant materials (200 g) were extracted thrice with 70% methanol (1 L) through cold maceration for 72 h at room temperature. The extracts were decanted and filtered through Whatman filter paper (No. 4, pore size of 20–25 µm). Using a rotary evaporator, the solvents were evaporated in vacuo at 40 °C. The extracts were then kept at −20 °C until further use.

### 4.4. Determination of Total Phenolic Content

Total phenol content was measured on a 96-well microplate and based on the Folin–Ciocalteu method described previously with some minor modifications [64]. A 100 µL volume of Folin–Ciocalteau reagent was mixed with 25 µL of GHE and DHE in a 96-well flat-bottomed microplate. The mixture was then shaken for 60 s. After leaving the mixture for 4 min, 75 µL of sodium carbonate (100 mg/mL) was added. After continuous shaking at mild speed for 1 min, incubation was carried out at room temperature for 2 h. Finally, the resulting colorimetric reaction was measured at 765 nm and compared with a standard curve generated fromgallic acid standard solutions. Gallic acid dilutions (3–500 ug/mL) wereused as a calibration standard, and total phenol content was expressed as milligrams of gallic acid equivalent (GAE) per gram of dry weight of plant extract (mgGAE/g).

### 4.5. Determination of Total Flavonoid Content

Total flavonoid content was determined by aluminum chloride colorimetric assay with slight modifications [65]. Standard concentrations of quercetin were prepared in 96% ethanol. A 50 µL volume of extract (1 mg/mL) or standard solution was added to 10 µL of 10% aluminum chloride solution, followed by 150 µL of 96% ethanol. A 10 µL volume of 1 M sodium acetate was further added to the mixture. Then, 96% ethanol was used as the reagent blank. All reagents were mixed and incubated for 40 min at room temperature in the dark. Absorbance was measured at 415 nm with a microplate reader, Varioskan Lux (Life Technologies Holdings Pte Ltd. Singapore 739256). TFC was expressed as milligrams quercetin equivalent (QE) per gram dry weight of plant extracts (mgQE/g).

### 4.6. Maintenance of Cell

The colon cancer cell line HCT 116 was grown in DMEM with 10% FBS and 1% Pen Strep and kept at 37 °C in humidified air with 5% CO_2_ in a CO_2_ incubator.

### 4.7. Qualitative Phytochemical Screening

The qualitative screening of phytochemicals was carried out using standard methods. The presence of alkaloids, terpenes, saponins, carbohydrates, steroids, tannins, phenols, flavonoids, and acids in the fruit extracts was determined [66].

### 4.8. GC-MS Analysis

GC-MS analysis of the *T. chebula* extracts was performed using gas chromatography (Trace 1300) integrated with mass spectrometry (TSQDuo) (Thermo Fisher Scientific (Waltham, MA, USA) Pvt Ltd., Blk 33 Marsiling Industrial Estate Road 3#07-06, Singapore–739256). A TG5 MS silica capillary column of 30 m × 0.25 mm i.d. and 0.25 µm film thickness was used for chromatography. The column oven temperature was initially maintained at 40 °C for 1 min and gradually increased with a 5 °C/m heating ramp and then maintained at 250 °C for 20 min. The injector and mass transfer line temperatures were set at 250 and 250 °C. An electron ionization energy system with an ionization energy of 70 eV was used for GC-MS detection. Helium gas was used as the carrier gas injected at a flow rate of 1 mL/min for each sample (1:100 extract in acetonitrile) with an injection volume of 0.5 µL. A 0.45 µM syringe filter was used to filter the dried extracts after they had been redissolved in the appropriate solvents. A 0.5 µL volume of the sample was then injected for GC-MS analysis. The highest reverse search index (RSI) value of mass spectra in the National Institute of Standards and Technology (NIST) GC-MS libraries was used to identify the compounds [67].

### 4.9. Quantitative Estimation of Gallic Acid and Ellagic Acid by HPTLC

Precoated aluminum-backed TLC sheets (20 × 10 cm) precoated with silica gel 60 F_254_(Merck, Mumbai, India)were used as the stationary phase. Ellagic and gallic acid solutions as standards (1 mg/mL) were prepared in HPLC-grade methanol. The standard solutionswere further applied on the HPTLC plate toprepare the standard plot. TheCAMAT Linomat V sample applicator (CAMAG, Muttenz, Switzerland) equipped with a 100 µL microsyringe (Hamilton, Bonaduz, Switzerland) under a flow of N_2_ (Nitrogen) Gas was used to load samples onto the TLC plates. For the development of the chromatogram, toluene–ethylacetate–formic acid–methanol in the ratio of 6:3:0.1:0.9 was optimized and used as the mobile phase for gallic acid quantification, while toluene–ethylacetate–formic acid in the ratio of 7:3:1 was used as the mobile phase for ellagic acid quantification. Samples were loaded onto the TLC plates, developed in a twin-trough chamber saturated with the mobile phase 20 min before plate development. and run up to 8 cm. The developed plates were dried and scanned at 254 nm using the CAMAG TLC Scanner IV, which was linked to visionCATS 3.0 software. Documentation of the TLC plates wasperformedusing CAMAG TLC Visualizer 2 under white, 254 nm, and 366 nm before and after derivatization with anisaldehyde solution. The software visionCATS by CAMAG was used to analyze the results [68].

### 4.10. Antioxidant Activity

#### 4.10.1. DPPH Free Radical Scavenging Assay

The ability of the extracts to neutralize the activity of the 1,1-diphenyl-2-picrylhydrazyl (DPPH) compound’s free radicals was used to measure itsantioxidant capacity using previously described microdilution techniques in 96-well plates with slightmodifications [69,70,71]. A range of concentrations (0–500 µg/mL) of the fruit extracts were prepared in methanol, and 100 µL each of the extracts and a freshly prepared methanolic solution of DPPH (150 µM) was incubated at 37 °C for 30 min. Afterwards, the absorbance was read in a microplate reader at 517 nm. The extracts were examined in triplicate, and the amount of DPPH radical scavenging activity was measured as a percentage versus a blank using the formula
% scavenging activity = (AC − AS/AC) × 100(1)
where AS is the absorbance of the sample or the standard, and AC is the absorbance of the control reaction (including all reagents but the test sample). Ascorbic acid and Trolox were used as standards. The IC_50_ value from the linear equation of the dose inhibition curve was obtained by plotting the extract concentration versus the corresponding percentage scavenging activity using GraphPad Prism 8.4.3. Results obtained are expressed as mean ± SD for experiments performedin triplicate.

#### 4.10.2. ABTS Free Radical Scavenging Assay

The extracts’ capacity to neutralize free radicals was also assessed using the ABTS cation decolorization assay [72,73]. A previously formed monocation of 2,2′-azino-bis (3-ethylbenzothiazoline-6-sulfonic acid) (ABTS •+) was reduced in the presence of a hydrogen-donating antioxidant. A cationic radical was prepared by reaction between 7 mM ABTS in water and 2.45 mM potassium persulfate (1:1), which was then kept at room temperature in the dark for 12 to 16 h before use. Methanol was then used to dilute the cationic solution prepared until an absorbance of 0.700 at 734 nm was reached. A range of concentrations (0–1000 µg/mL) of the plant extracts were prepared in methanol, and a known volume of the extract was made to react with an equal volume of the diluted ABTS•+ solution. The absorbance was measured after 30 min of incubation in the dark following an initial mixing. Each experiment was performedwith a suitable solvent blank. All experiments were carried out in triplicate, and the percentage inhibition of absorbance at 730 nm was calculated using the formula
ABTS•+ scavenging effect (%) = (AC − AS/AC) × 100 (2)
where AS is the absorbance of the sample/standard, and AC is the absorbance of the control reaction (including all reagents but the test sample). Ascorbic acid and Trolox were used as standards [74]. The IC_50_ values were determined in a manner identical to that described for the DPPH assay, and were compared to that of the standards Trolox and ascorbic acid.

### 4.11. Cell Viability Assay

The 3-(4,5-dimethylthiazol-2-yl)-2,5-diphenyltetrazolium bromide (MTT) assay, described by Mosmann, was used to assess the anticancer effect of the extracts [75]. Cultured cells were harvested, and 5 × 10^3^ cells were seeded in a 96-well plate (final volumes of 100 µL per well) and further incubated for 24 h at 37 °C in a CO_2_ incubator. Increasing concentrations (0–500 µg/mL) of the extracts were treated on the cells and further incubated for 48 h. A 15 µL volume of MTT solution (5 mg/mL) was added to each of the 96-well plates and incubated for 4 h at 37 °C. A 100 µg/mL amount of DMSO was added to each well to solubilize the water-insoluble purple formazan crystals formed. Absorbance was read at 570 nm using a microplate reader (Varioskan LUX multimode microplate reader, ESW version 1.00.38) from Thermo Fisher Scientific (Waltham, MA, USA). The generated dosage response curve was used to calculate the IC_50_ and the percentage of cell growth inhibition.

### 4.12. Colony-Forming Assay

The effect on the clonogenicity of the HCT 116 cells was determined by colony-forming assay as described previously [35]. In a 6-well culture plate, a total of 1000 cells suspended in 1.5 mL of culture medium were sown in each well. After incubating for 24 h, fresh media containing different concentrations of the extracts were added, replacing the old medium. Following an incubation time of 48 h, the media containing the different concentrations of the extract were removed, the cells were washed with sterile PBS, and fresh media were added toeach well. The cells were incubated for approximately 12 days undisturbed to allow colony formation. After the incubation process, the cells were fixed with 4% paraformaldehyde and stained with 0.5% crystal violet solution, washed with distilled water, air-dried, and colony numbers were counted.

### 4.13. Test for Cytotoxicity Using LDH Assay

Lactate dehydrogenase(LDH) is a cytosolic enzyme released into the cell and the surrounding cell culture media upon damage to the plasma membrane. An enzyme-coupled reaction where LDH catalyzes the conversion of lactate to pyruvate via NAD+ to NADH allows for the quantification of the extracellular LDH released into the medium. Oxidation of NADH by diaphorase leads to the reduction of tetrazolium salt to a red formazan product that can be measured spectrophotometrically at 490 nm. The amount of LDH released into the medium is directly related to the level of formazan production, which denotes cytotoxicity [76]. CyQuant LDH Cytotoxicity Assay Kit (Invitrogen) was used for determining the percentage cytotoxicity using the formula
% cytotoxicity = (Compound treated LDH activity − Spontaneous LDH activity/Maximum LDH activity − Spontaneous LDHactivity) × 100(3)

### 4.14. Effect on Cell Migration

Using the wound-healing experiment previously developed, the impact on the migratory capacity of HCT 116 was investigated [77]. Single suspensions of cells were prepared in culture media, and 3×10^5^ cells were seeded per well of a 6-well culture plate. After 48 h of incubation, when the cells’ growth has finally reached 100% confluency, a scratch was made in the middle of each well with a 200 µL pipette tip. The edges formed due to the scratch were smoothened using culture media. Afterwards, a treatment containing the hydroalcoholic extract in increasing concentrations was added to each well with a final volume of 2 mL. The scratches were photographed for zero hours (0 h), twenty-fourhours (24 h), and forty-eight hours (48 h) at 10×magnification in an inverted light microscope. About 50 points per well were used to measure the width of the cell-free area, and the average percentage of wound closure was calculated relative to zero time.

### 4.15. Giemsa Staining

HCT-116 cells in the logarithmic growth phase were plated in 6-well cultures and kept for 24 h at 37 °C with 5% CO_2_. After the incubation process, cells were treated with increasing concentrations(30, 60, and 80 µg/mL) the extract or standard drug (5-FU) and kept for 48 h of incubation. After the treatment, the cells were washed with PBS and fixed with methanol. The cells were then stained with 6% Giemsa solution for 5 min, washed with distilled water to remove excess stain, and dried. Changes in the morphology of the cells were observed using an inverted light microscope [78].

### 4.16. DAPI Staining

HCT116 cells (1.5 × 10^5^) were seeded in a 6-well culture plate with sterile coverslips previously treated with fibronectin. Incubation at 37 °C with 5% CO_2_ for 24 h was followed by treatment of the cells with increasing concentrations of TYH and standard drug 5-FU. Cells were then incubated for 48 h. Afterward, the culture was discarded, and the cells were washed with PBS twice and fixed with 4% paraformaldehyde for 10 min. After that, the appropriate amount of DAPI (1 mg/mL; Invitrogen, USA) was added to cover the surface of the bottom of the well and allowed tostain in the dark for 5 m [78]. Finally, the glass coverslip was mounted on a glass slide, and nuclear morphology was observed using a fluorescence microscope, Nikon Eclipse Ni-U (Nikon Corporation, Tokyo, Japan) equipped with camera, Nikon DS-Ri2 (Nikon Corporation, Tokyo, Japan).

### 4.17. Caspase-3 Activity Assay

Caspases are essential apoptotic or programmed cell death mediators. Caspase-3 is one of the numerous caspases and is essential for DNA fragmentation, chromatin condensation, and other markers of apoptosis. It is a frequently activated protease in mammalian cell apoptosis [79]. The EnzChek^®^ Caspase-3 Assay Kit #1 (Thermo Fisher Scientific, Waltham, MA, USA) was used to evaluate the activation of caspase-3 in HCT 116 cells after treatment with TYH. By monitoring changes in caspase-3 and other DEVD-specific protease activity, the EnzChek^®^ Caspase-3 Assay Kit #1 enables the detection of apoptosis [80]. Cells were seeded in 6-well culture plates and treated with increasing concentrations (10, 20, and 30 µg/mL) of TYH for 48 h. After that, cells were washed with ice-cold PBS and processed as per the protocol given by the manufacturer. The 7-amino-4-methylcoumarin-derived substrate Z-DEVD-AMC, where Z stands for a benzyloxycarbonyl group, forms the basis of the test. This substrate is cleaved by proteases to produce a vivid blue fluorescent product (excitation/emission 342/441 nm). The flourescence is measured using Varioskan Lux (Life Technologies Holdings Pte Ltd. Singapore 739256). The kit allows for the continuous monitoring of caspase-3 activity and that of closely related proteases in cell extracts. The fluorescence intensity is proportional to caspase-3 activity. The amount of AMC released in the reaction was quantified using the reference standard 7AMC.

## 5. Statistical Analysis

GraphPad Prism 8.4.3 was used for the statistical analysis and representation of all raw data obtained. Significant differences in the treatment groups were determined using a one-way analysis of variance (ANOVA), followed by Tukey’s multiple comparison test at the 5% level (*p* < 0.05).

## 6. Conclusions

The hydroalcoholic extracts of *T. chebula* fruit specimens collected from different locations in Manipur were found to be rich in phenolics and tannins. A significant variation in the extracts’ phytochemical composition was observed, indicating the diversity of *T. chebula* found in Manipur. The extracts were also found to possess potent antioxidant capacity compared to the standards ascorbic acid and Trolox as determined by DPPH and ABTS assays. The extracts were found to possess significant cytotoxicity against HCT-116 colon cancer cells. They inhibited cell proliferation, colony-forming ability, and migration of HCT 116 cells in a dose-dependent manner, accompanied by a significant activation of caspase-3 enzyme activity. All these findings indicate the promising anticancer activity of the hydroalcoholic extract of *T. chebula* in colon cancer cells.

## Figures and Tables

**Figure 1 molecules-28-02901-f001:**
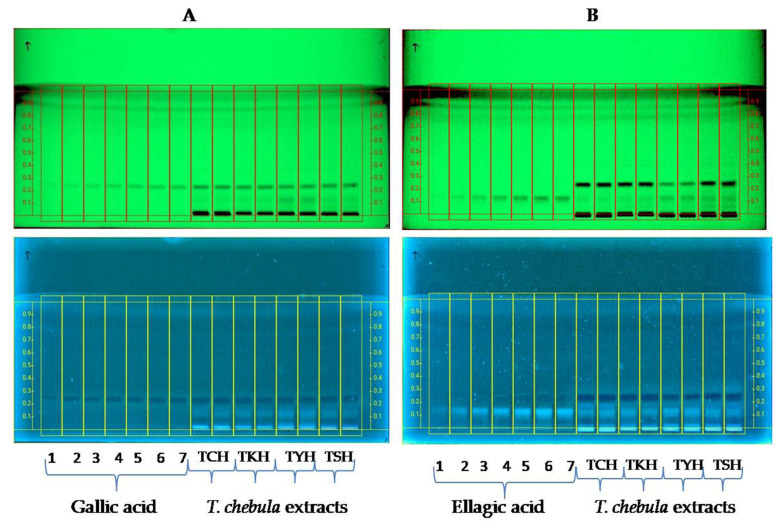
HPTLC chromatogram of increasing concentrations of gallic acid (**A**) and ellagic (**B**) and hydroalcoholic extracts of *T. chebula* fruits (TCH, TKH, TYH, and TSH) at 254 nm and 366 nm.

**Figure 2 molecules-28-02901-f002:**
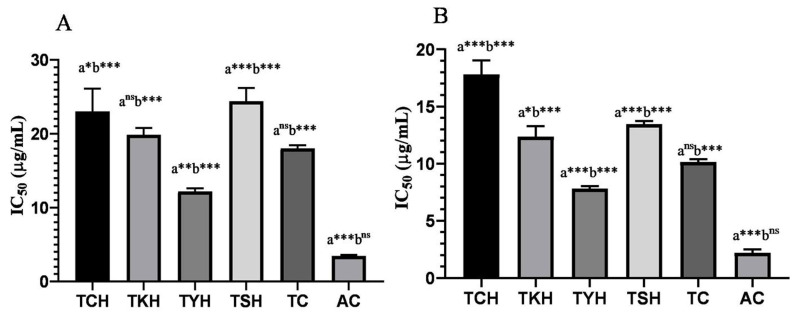
Antioxidant activity of *T. chebula* extracts from different locations in Manipur., standards Trolox (TC) and ascorbic acid (AC) as determined by (**A**) DPPH assay and (**B**) ABTS assay. Values represent mean ± SD of three independent experiments. Statistical significance was set at *p* < 0.05. (*** *p* < 0.001; ** *p* < 0.01; * *p* < 0.05) One-way ANOVA was carried out for statistical comparison (‘a’, comparison with TC; ‘b’, comparison with AC).

**Figure 3 molecules-28-02901-f003:**
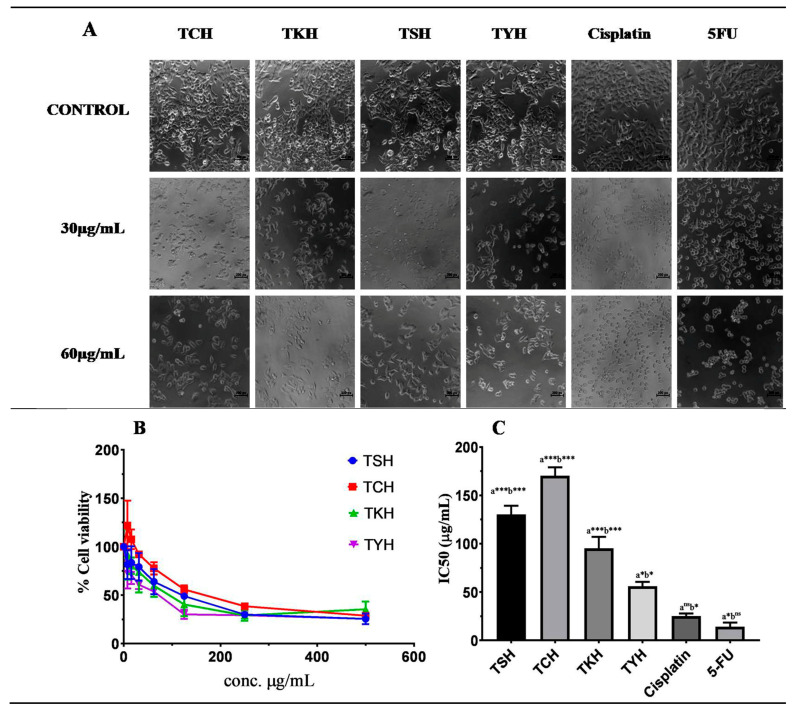
(**A**) Morphology of cells as observed, under inverted light microscope, after treatment with varying concentrations (0, 30, and 60 µg/mL) of *T. chebula* extracts for 48 h. (**B**) Cytotoxic effect of *T. chebula* extracts on HCT116 cells was determined using MTT assay. Cells were seeded in 96-well plates and treated with varying concentrations (0–500 µg/mL) of extracts. MTT assay was performed after 48 h. (**C**) Graphical representation of IC_50_ values of *T. chebula* extracts and standard drugs cisplatin and 5-FU. One-way ANOVA; *p* < 0.05; *n* = 3 (*** *p* < 0.001; * *p* < 0.05) (“a”, comparison with cisplatin; “b”, comparison with 5-FU).

**Figure 4 molecules-28-02901-f004:**
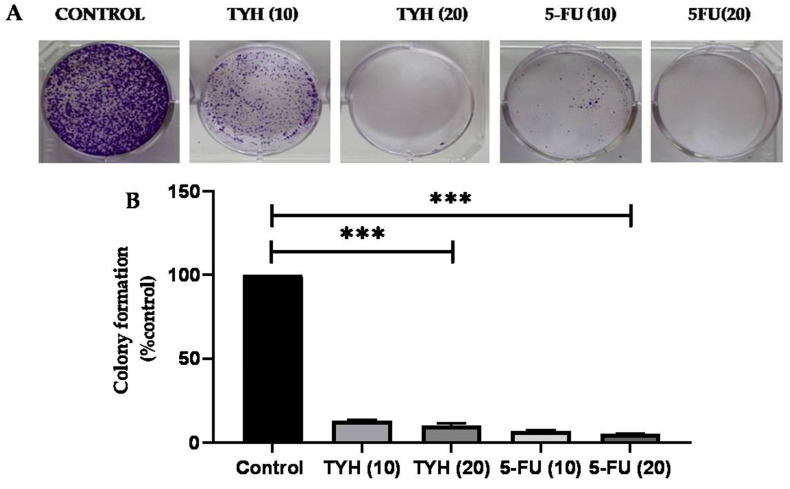
(**A**) Cells’ capacity to form colonies after being exposed to various doses of TYH for 24 h. Crystal violet dye was used to stain cells after they had been fixed with 4% paraformaldehyde, and a colony counter was used to count the number of colonies. (**B**) Graph showing % colonies formed after treatment with different concentrations (0, 10, and 20 µg/mL) of TYH for 24 h. The values indicate mean ± SD (*n* = 3), *** *p* < 0.001 vs. control. Graph shows dose-dependent decrease in % colony formation relative to control.

**Figure 5 molecules-28-02901-f005:**
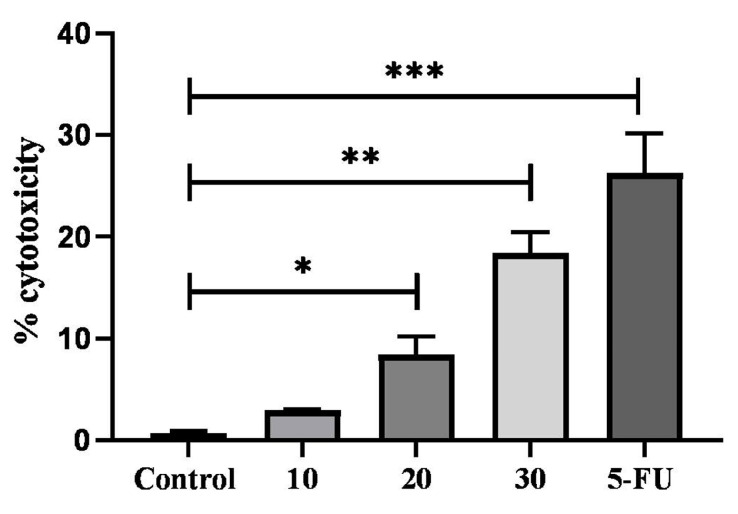
HCT-116 cells were treated with increasing concentrations (10, 20, and 30 µg/mL) of TYH. 5-FU (10 µg/mL) was used as positive control. Incubation was carried out for 48 h. Release of LDH was detected using CyQUANT™LDH Cytotoxicity Assay Kit. Data are represented as mean ± SD (*n* = 3), (*** *p* < 0.001; ** *p* < 0.01; * *p* < 0.05) vs. control.

**Figure 6 molecules-28-02901-f006:**
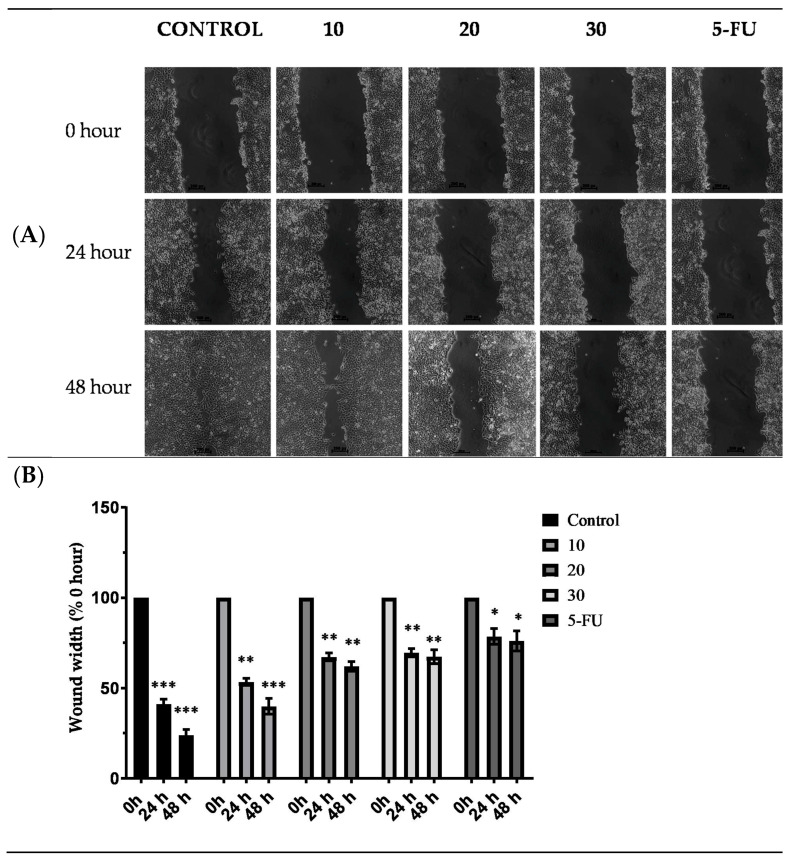
(**A**) TYH inhibited metastatic properties of HCT-116 cells. Inhibition of migration of HCT-116 cells after treatment with 10, 20, and 30 µg/mL of TYH extract and 5-FU (5 µg/mL) for 24 and 48 h. (**B**) Quantitative representation of migration of HCT-116 cells by wound-healing assay. Results are presented as percentage control at 0 h and represent mean ± SD values from three independent experiments. * *p* < 0.05; ** *p* < 0.01; *** *p* < 0.001 vs. 0 h.

**Figure 7 molecules-28-02901-f007:**
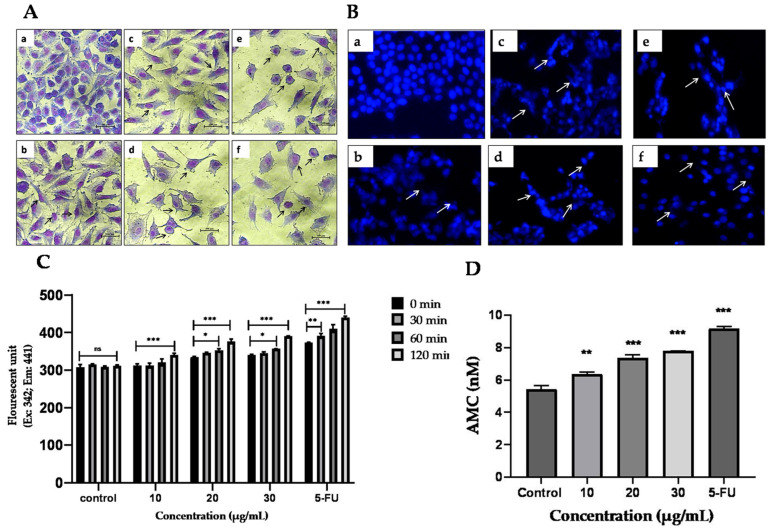
Morphological detection of apoptosis through Giemsa (**A**) and DAPI (**B**) staining. HCT 116 cells were treated with TYH and 5-FU for 48 h, (**a**) Control (non-treated), (**b**) 30 µg/mL, (**c**) 60 µg/mL, (**d**) 90 µg/mL, (**e**) 120 µg/mL, and (**f**) 5-FU (10 µg/mL) and changes in cell morphology were examined. Further effect of TYH on caspase-3 enzyme activity in HCT 116 cells after 48 h of treatment was checked using EnzChek^®^ Caspase-3 Assay Kit. (**C**). Time-dependent increase in caspase-3 enzyme activity after treatment with TYH for 48 h. (**D**) Quantification of AMC release in cells treated with TYH after an incubation time of 120 m. Results represent mean ± SD (*n* = 3), where *** *p* < 0.001; ** *p* < 0.01; * *p* < 0.05 vs. control.

**Table 1 molecules-28-02901-t001:** Sample name, collection site, % yield of extracts, andcontent (µg/mg) of gallic and ellagic acid determined by HPTLC.

Sample	Collection Site with GPS Coordinates	Yield Percentage (*w*/*w*)	Gallic Acid Content (µg/mg) as Determined by HPTLC	Ellagic Acid Content (µg/mg) as Determined by HPTLC
TCH	Chandel, N 24°25′35.7″ E94°00′47.7″	25.2	22.44 ± 0.056	2.652 ± 0.07
TSH	Sagolband, N 24°48′06″ E93°55′16″	28.7	21.61 ± 0.183	3.597 ± 0.145
TKH	Kakching, N 24°30′01.9″ E93°58′27.1″	30.3	18.275 ± 0.272	3.081 ± 0.089
TYH	Yumnam Huidrom, N 24°39′46″ E93°54′14″	28.7	19.629 ± 0.055	11.265 ± 0.089

**Table 2 molecules-28-02901-t002:** Chemical composition of hydroalcoholic extracts of *T. chebula* collected from different locations inManipur.

Sl.No.	Name of the Compounds	% Content in TCH	% Content in TKH	% Content in TYH	% Content in TSH
1	1,2,3-Benzenetriol	20.95	26.53	43.56	40.41
2	2-Cyclopenten-1-one	19.35	12.79	9.07	14.66
3	Catechol	12.82	6.85	NA	12.47
4	5-Hydroxymethylfurfural	6.04	11.26	4.03	NA
5	1-Hexyl-2-nitrocyclohexane	4.13	2.76	NA	2.11
6	1-Undecanol	3.15	1.88	NA	1.74
7	Phenol	3.27	0.37	5.40	NA
8	Hexadecanamide	1.95	1.31	NA	1.56
7	4H-Pyran-4-one, 2,3-dihydro-3,5-dihydroxy-6-methyl	1.74	1.68	NA	0.74
8	2,6-Difluorobenzoic acid, 4-nitrophenyl ester	1.72	3.05	NA	NA
9	1-(1′-Pyrrolidinyl)-2-butanone	1.33	NA	NA	NA
10	Phosphonic acid, (p-hydroxyphenyl)-	NA	3.82	NA	4.64
11	Sucrose	0.22	2.27	NA	NA
12	3-Furaldehyde	0.68	1.59	0.89	0.50
13	2-Cyclohexen-1-one	NA	1.49	NA	1.22
14	Cyclobutaneethanol, á-methylene	0.92	1.40	0.93	NA
15	Benzeneacetaldehyde	0.89	1.13	1.56	NA
16	2,3,4,5-Tetrahydropyridazine	NA	NA	1.65	NA
17	1,2-Benzenediol, mono(methylcarbamate)	NA	0.19	1.57	NA
18	2,5-Furandione, 3-methyl-	NA	NA	1.18	0.18
19	2,6,10,14-Tetramethylpentade can-6-ol	NA	NA	1.06	NA
20	S-Methyl 2-methylpropanethioate	NA	NA	NA	1.26

**Table 3 molecules-28-02901-t003:** Antioxidant activity of four *T. chebula* fruits from different geographical regions of Manipur, India.

Sample	DPPH Assay IC_50_ (µg/mL)	ABTS Assay IC_50_ (µg/mL)	TPC (GAEmg/g ext)	TFC (QEmg/g ext.)
TCH	23.00 ± 3.11	17.81 ± 1.21	286.76 ± 0.35	8.98 ± 0.35
TKH	19.85 ± 0.93	12.35 ± 0.92	295.35 ± 0.98	8.10 ± 0.87
TYH	12.16 ± 0.42	7.80 ± 0.23	304.56 ± 1.23	7.65 ± 0.45
TSH	24.42 ± 1.78	13.45 ± 0.28	270.65 ± 0.78	7.95 ± 0.79
Trolox(TC)	18.01 ± 0.44	10.15 ± 0.24	-	-
Ascorbic Acid(AC)	3.46 ± 0.12	2.19 ± 0.003	-	-

**Table 4 molecules-28-02901-t004:** IC_50_ values of *T. chebula* extracts in HCT 116 cells.

Sample/Drugs	IC_50_ (µg/mL) Values of the *T chebula* Extracts against HCT 116 Cells
TSH	133.58 ± 2.0
TCH	172.05 ± 2.0
TYH	52.42 ± 0.87
TKH	94.07 ± 1.70
Cisplatin	24.38 ± 1.26
5-FU	12.75 ± 1.03

**Table 5 molecules-28-02901-t005:** Details of sample collection sites.

Sl.No	Sample (Place of Collection)	Voucher Specimen No.	Latitude Longitude	Elevation(m)
1	TCH (Chandel, Manipur)	IBSD/M-283-A	N 24°25′35.7″ E94°00′47.7″	817
2	TKH (Kakching Manipur)	IBSD/M-283-C	N 24°30′01.9″ E93°58′27.1″	778
3	TSH (Sagolband, Imphal, Manipur)	IBSD/M-283-B	N 24°48′06″ E93°55′16′	780
4	TYH (Yumnam Huidrom, Imphal, Manipur)	IBSD/M-283-D	N 24°39′46″ E93°54′14″	784

## Data Availability

Data are available within the manuscript.

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
