# Peer review of "Chemical Profiling and Therapeutic Evaluation of Standardized Hydroalcoholic Extracts of *Terminalia chebula* Fruits Collected from Different Locations in Manipur against Colorectal Cancer"

_molecules, 2023, doi:10.3390/molecules28072901_

Round 1
Reviewer 1 Report
1. What new active constituents the authors had found in the extracts?
2. I think the statistics of colorectal cancer (line 378-385) should place in the introduction section.
3. What are the significance to add cytotoxicity against HCT116, as the activity is very less as compared to the standard drugs (Cisplatin and 5-FU)?
4. Rather than reporting hydroalcoholic extract, the authors should co-relate which active constituents produced significant antioxidant and cytotoxicity.
Reviewer 2 Report
In this report, the authors isolated and characterized the major chemical components present in Terminalia chebula fruits collected from different locations of Manipur and evaluated the therapeutic potential against colorectal cancer and found effective results. This type of natural remedies will be always important due to lesser side effects than synthetic drugs.
This work is highly relevant and important to the readers of this journal due to following reasons:
Terminalia chebula fruits are already regarded as the herbal medicines in India for various diseases and this work unravels the chemical constituents present in the fruit. Further, colorectal cancer is a chronic disease for modern society. So, if the natural extracts of Terminalia chebula fruits could cure it, it may be the alternative therapy as compared to the existing therapies.
My comments are as follows:
1. The title and the abstract of the paper is presented properly and reflect the objective of the work.
2. The experiments performed for analysis and anticancer evaluation are found to be rational.
3. The satisfactory results are found, for eg. The IC50 value against HCT 116 (TYH) is better as compared to the reference standard.
4. The major compounds found are 2-Cyclopenten-1-one, 5-Hydroxymethylfurfural and Catechol and since the extract is a mixture of all those compounds, it is difficult to know the exact molecule responsible for anticancer activity? authors may comment on it. If so, are there any possible comparative study to address this issue?
5. Are there any possible side effects of these compounds reported yet?
6. The structures of all compounds may be presented in the discussion section.
7. The images and tables are presented correctly (with proper numbering) in line with the text presented.
Following typo errors needs to be fixed;
a. CO2 should be written properly with suffix.
b. Please check the formatting issue for X mg/mL and X h.
c. Please use the "space" properly.
8. Authors are suggested to cite ‘’Org. Lett. 2018, 20, 9, 2611–2614’’ as one of the chemical methods for synthesis of Tumor-associated antigens in the introduction. In my opinion this work is very suited for publication, so after addressing the points mentioned above, I recommend this article to publish in molecules.
Reviewer 3 Report
Comments to the Author
The new product from natural chemistry is required due to the advancement of human use. This paper fits well with this theme and discusses the possibilities of using natural plants for this purpose, but it needs reworking. The experiment is not clearly presented and not statistically representative. In particular:
Introduction would be through, with references. All abbreviations, they have been expanded in text and same have been removed from the abbreviation list. Please note that we were unable to renumber the reference citations because some references were not cited in the article.
Material and methods described more clearly and explained in more details:
- plant materials: number, replications (it is necessary to have more plants of the same species and the same portion).
Results and discussion could be improved.
References do not meet the indication of the journal.
Specific comments;
- Some abbreviations, such as GC-MS, should be revised.
- With the data varieties of T. chebura, Table 1 may contain inaccuracies about tannin and polyphenol that should be reconsidered.
- Antioxidant activity of four T. chebula in Table 3 could be statistical comparison.
- In text citation, some of those could be improved as the indication of the journal, such Liang et al.
- The methodology for statistical hypothesis testing must be mentioned.
Reviewer 4 Report
The article to determine that exploring the phytochemical profile of the hydroalcoholic extract of T. chebula fruits collected from different locations of Manipur, India and to evaluate the anticancer activity against colon cancer cell HCT 116. GCMS was used for identifying the volatile constituents of the extracts and quantifying the relative abundance of each constituents in the extract. HPTLC analysis is a techniques which is commonly used for validation of traditional medicines derived from plants. Two major hydrolysable tannins, gallic and ellagic acid were also quantified from the fruit extracts of T. chebula using HPTLC. DPPH and ABTS assays were used to measure the extracts' antioxidant properties. The anticancer activity of T. chebula extract was studied in HCT 116 colon cancer cells and mechanism for cell cytotoxicity was studied using various. The topic is interesting, the manuscript as a whole is well-organized, well-written, and updated. I have some suggestion to improve the readership of manuscript.
Minor comments:
▪ In abstract I think you need to add some data from the Aarticle.
▪ Number of reference not enough in Introduction (17 refrances only??)
▪ Some of the material and methods sections have not clear descriptions. Please, check it and improve the grammatical English (please, use correct verb tenses).
▪ The manuscript has grammatical errors and needs improvement.
▪ On overall, Correlation of chemical pattern from plant associated with total phenol and flavonoids can be very helpful (supported by the findings of other authors).
▪ The authors should add the symbol of significant difference of statistical analysis in all graphs.
▪ What is the difference between cell viability and cell proliferation after assayed assay? The authors should explain how anticancer can determine cell proliferation in this study.
• The conclusion is too general, it should be connected and supported with the results.
• please revise all name of plant in the text and change it to italic
• Please check typo in whole text.
• The design methodology of experiments was professionally organized. The results obtained are interesting.
• The abbreviations of some journals in References are incorrect
• References: Please update by 2022 and 2023 References.
• References not in the format of the journal.
• I recommend that the Editorial office consider this manuscript for publication after minor revision.
Round 2
Reviewer 1 Report
Manuscript is ok now
Reviewer 3 Report
Based on my review report, the authors evaluated the chemical content of Terminalia chebula fruits and identified potentially significant chemical components present in the collection from various place. Additionally, they obtained and described these significant chemical elements. I therefore have nothing more to suggest.